# Targeting the NO–sGC–cGMP Pathway: Mechanisms of Action of Vericiguat in Chronic Heart Failure

**DOI:** 10.3390/cells14171400

**Published:** 2025-09-08

**Authors:** Tine Bajec, Gregor Poglajen

**Affiliations:** 1Department of Internal Medicine, University Medical Centre, 1000 Ljubljana, Slovenia; tinebajec1@gmail.com; 2Advanced Heart Failure and Transplantation Center, Department of Cardiology, University Medical Centre, 1000 Ljubljana, Slovenia

**Keywords:** heart failure, soluble guanylate cyclase stimulators, vericiguat

## Abstract

The recent advancements in the medical management of patients with chronic heart failure with reduced ejection fraction (HFrEF) is the soluble guanylate cyclase (sGC) stimulator, vericiguat. Clinical trials have demonstrated that vericiguat effectively lowers plasma levels of NT-proBNP and reduces the risk of cardiovascular death or hospitalization in HFrEF patients, making it a class IIb recommendation for patients with worsening heart failure despite receiving guideline-directed medical therapy. However, the precise pathophysiological mechanisms underlying these clinical benefits remain unexplored. This review aims to present the signalling pathways associated with maladaptive remodeling and heart failure progression that can be modulated by sGC stimulators, focusing on the antihypertrophic, antifibrotic, and anti-inflammatory effects of NO–sGC–cGMP signalling observed in preclinical studies. A better understanding of the mechanisms of action of sGC stimulators could optimize heart failure treatment strategies and enable tailoring of therapies to individual patient profiles.

## 1. Introduction

Heart failure (HF) represents a significant global health burden, affecting more than 60 million people worldwide [1]. Its prevalence is expected to rise further due to population aging, increasing rates of metabolic syndrome–associated comorbidities, and improved survival following myocardial infarction [2,3]. Despite therapeutic advances, the one-year mortality rate of chronic HF remains above 5%, underscoring the need for new treatment strategies [1].

In HF, reduced cardiac output activates the renin–angiotensin–aldosterone system (RAAS) and sympathetic nervous system to preserve peripheral perfusion. While initially compensatory, chronic activation promotes maladaptive myocardial remodeling, accelerates disease progression, and contributes to recurrent episodes of clinical worsening that often require intravenous diuretics or hospitalization [4]. Guideline-directed therapy with β-blockers, mineralocorticoid receptor antagonists (MRAs), and angiotensin-converting enzyme inhibitors (ACEIs) or angiotensin receptor blockers (ARBs) interrupts this neurohumorally driven maladaptive cycle. The PARADIGM-HF and DAPA-HF trials further expanded the treatment landscape with the introduction of angiotensin receptor–neprilysin inhibitors (ARNIs) and sodium–glucose cotransporter 2 (SGLT2) inhibitors [5,6]. While SGLT2 inhibitors exert multiple pleiotropic effects, ARNIs have a more clearly defined mechanism: in addition to RAAS modulation, neprilysin inhibition prolongs the half-life of natriuretic peptides, which, via particulate guanylate cyclase (pGC), increase cGMP production. Elevated cGMP activates signalling pathways that promote diuresis, natriuresis, and vasodilation, ultimately alleviating congestion and reducing cardiac workload [7].

Despite optimal GDMT, the risk of cardiovascular death and HF hospitalization remains high in contemporary HF trials, suggesting that additional pathogenic pathways not addressed by current therapies contribute to disease progression [5,6,8]. One such pathway is NO–sGC–cGMP signalling, which is markedly impaired in HF due to heightened oxidative stress and systemic inflammation (Figure 1). Reduced NO synthesis and bioavailability diminish sGC activation, while oxidative modification or loss of the sGC haem group renders the enzyme insensitive to NO [9,10]. Concurrently, phosphodiesterase (PDE) upregulation accelerates cGMP degradation, further weakening this cardioprotective cascade [10]. The resulting cGMP deficit disrupts autophagy and proteasomal function in cardiomyocytes and promotes hypertrophic and fibrotic signalling, thereby exacerbating maladaptive remodelling [9,10,11].

While the beneficial effects of ARNIs are partly mediated through increased cGMP generation, this occurs via membrane-bound pGC rather than cytosolic sGC. Because of these differences in subcellular localization, pGC and sGC generate cGMP in distinct intracellular compartments, leading to activation of separate downstream signalling pathways. This supports NO–sGC–cGMP as a mechanistically distinct therapeutic target [12].

Consequently, there is a strong rationale for pharmacological augmentation of the NO–sGC–cGMP signalling pathway to restore its cardioprotective effects in heart failure patients. This may be particularly advantageous in patients with recent worsening heart failure, where oxidative stress and inflammation are amplified, further compromising the generation of cGMP. Importantly, this subgroup remains at high residual risk on optimal GDMT with reported event rates of up to 16%, underlining the persistent limitations of existing therapies in such patients [8,13].

sGC stimulators represent a novel approach for pharmacological stimulation of the NO–sGC–cGMP signalling pathway. By binding to sGC, these stimulators act as positive allosteric modulators to increase its affinity for NO and partially activate sGC even in the absence of NO [14]. Due to their dual mechanism of action, sGC stimulators can enhance cGMP production even in conditions of low NO bioavailability, which is commonly seen in heart failure.

Clinical trials have demonstrated that the sGC stimulator vericiguat can lower plasma levels of NT-proBNP and reduce the risk of cardiovascular death or hospitalization in high-risk patients with HFrEF [15,16]. More recently, the VICTOR trial extended these findings to a stable HFrEF population. The trial did not meet its primary composite outcome of cardiovascular death or HF hospitalization; however, it demonstrated significant reductions in both cardiovascular and all-cause mortality [17]. The precise pathophysiological mechanisms underlying these clinical benefits remain to be fully elucidated.

The aim of this review is to provide an overview of the current knowledge of the signalling pathways involved in pathological cardiac remodeling and inflammation in heart failure that, based on preclinical data, can be modulated by sGC stimulators. The effects of NO–sGC–cGMP stimulation are categorized as antifibrotic, antihypertrophic, and anti-inflammatory. However, since pathological remodeling and inflammation are closely interrelated, this categorization is somewhat arbitrary, as most of the pathways discussed influence all three processes. In addition, key clinical trial data on vericiguat in chronic heart failure are briefly summarized to assess how well the mechanisms described in the preclinical studies align with clinical outcomes.

## 2. Antifibrotic Effects

The extracellular matrix (ECM) is crucial for myocardial function, providing structural support to cardiomyocytes and aiding signal transmission [18]. Hence, the composition of cardiac ECM is a precisely regulated process, balancing ECM protein deposition with degradation mediated by metalloproteinases. In heart failure, chronic mechanical stress and neurohormonal dysregulation trigger the proliferation and activation of cardiac fibroblasts, leading to their differentiation into myofibroblasts [18]. Collectively, these remodelling processes lead to collagen accumulation within the myocardial interstitium, resulting in structural alterations known as cardiac fibrosis.

Increased fibrous content and collagen crosslinking in the ECM reduce myocardial compliance, raising left ventricular filling pressures [19]. Additionally, heightened internal friction within the collagen-rich interstitium diminishes early diastolic recoil of the ventricle, impairing early diastolic filling [19]. Beyond diastolic dysfunction, the spatial reorganization of the ECM decreases the efficiency of mechanical force transmission thereby compromising myocardial contractility [19]. This creates a positive feedback loop where cardiac fibrosis induces both diastolic and systolic dysfunction of the myocardium further promoting collagen deposition. Myocardial fibrosis additionally carries an increased risk of arrhythmias as it makes the substrate susceptible to reentry and disrupts the conduction system. Consequently, cardiac fibrosis is a significant contributor to morbidity and mortality in patients with chronic heart failure.

Preclinical studies show that sGC stimulators have potent antifibrotic effects, reducing fibrosis accumulation in multiple organs—including the kidneys, liver, dermis, lungs, and heart [20]. In Dahl salt-sensitive rats on a high-salt diet, treatment with riociguat, another sGC stimulator, resulted in lower plasma levels of profibrotic biomarkers (osteopontin, tissue inhibitor of matrix metalloproteinase-1 and plasminogen activator inhibitor-1) and decreased interstitial myocardial fibrosis compared to controls [21]. Similarly, in acute heart failure models, sGC stimulators reduced myocardial collagen and myofibroblast accumulation [22]. Importantly, these effects were independent of blood pressure, indicating that mechanisms beyond simple blood pressure reduction contribute to the observed antifibrotic outcomes [22]. Although there is solid evidence of sGC stimulators reducing cardiac fibrosis in animal models, the exact pathophysiological mechanisms and the extent of their contribution remain unclear.

## 3. TGF-β Signalling

Profibrotic processes in heart failure are mediated by various signalling molecules, including transforming growth factor beta (TGF-β). Chronic mechanical stress and neurohumoral activation cause cardiac fibroblasts, myocytes, and macrophages/monocytes to increase TGF-β expression. TGF-β then binds to activin receptor-like kinase (ALK) 5, initiating transmembrane signalling [23,24]. The intracellular domains of ALK 5 possess serine/threonine kinase activity, which is essential for activating a group of intracellular signalling molecules called Smads. After phosphorylation by ALK 5, Smad2 or Smad3 forms a heterodimer with Smad4 [23]. This complex translocates to the nucleus, where it acts—together with other signalling molecules—as a transcription factor. It regulates expression of fibrosis-associated proteins such as collagens, plasminogen activator inhibitor-1, proteoglycans, integrins, connective tissue growth factor, and matrix metalloproteinases [25]. Additionally, TGF-β increases the expression of an inhibitory Smad molecule, Smad7, which suppresses ALK 5-mediated activation of Smad2/3, providing an autoregulatory negative feedback loop [25]. In addition to Smad-dependent pathways, TGF-β transmits profibrotic signals to the cellular nucleus through Smad-independent mechanisms, such as the Mitogen-Activated Protein Kinase (MAPK)/Extracellular Signal-Regulated Kinase (ERK) pathway [26]. This activation begins at phosphorylated tyrosine residues on the ALK 5 intracellular domain. These sites recruit the ShcA-Grb2-Sos complex, which is essential for Ras activation [26]. Activated Ras then triggers a kinase cascade involving Raf, MEK1/2, and ultimately ERK1/2 [26,27]. Once activated, ERK1/2 translocates to the nucleus, where it can act as a transcription factor or regulate the activity of other transcription factors through phosphorylation and chromatin remodeling [27].

In mice renal fibroblasts, augmentation of NO–sGC–cGMP signalling, either through pretreatment with sGC stimulators or direct cGMP application, resulted in reduced nuclear translocation of phosphorylated Smad3 and decreased levels of phosphorylated ERK1/2 [28]. These effects were PKG (protein kinase G)-dependent, as no changes in Smad3 and ERK1/2 activity were observed in PKG-knockout kidneys [28]. Similarly, sGC stimulator treatment reduced ERK phosphorylation in human dermal fibroblasts. However, it did not affect Smad-mediated signalling in this setting [29]. In pulmonary artery smooth muscle cells, PKG phosphorylation of Smad3 facilitated its sequestration in the cytoplasm by increasing its affinity for the cytosolic anchoring protein β2-tubulin [30]. This binding of Smad3 to β2-tubulin was also observed in renal fibroblasts, although this process was cGMP-independent [28]. Additionally, in human hepatic stellate cells and animal aortic endothelial cells, activation of the NO–sGC–cGMP pathway exerted antifibrotic effects through different mechanisms: upregulation of inhibitory Smad7 and induction of Smad2 proteasomal degradation (Figure 2) [31,32]. These findings suggest that sGC stimulators regulate TGF-β signalling through a dual mechanism: Smad-dependent inhibition and Smad-independent inhibition, both of which are tissue-specific. Therefore, to better understand the potential antifibrotic effects of sGC stimulators in the heart, cardiac fibroblasts as drivers of cardiac fibrosis warrant separate discussion.

Studies exploring TGF-β signalling in mouse cardiac fibroblasts demonstrated that enhancing cGMP-PKG activity resulted in the phosphorylation and reduced nuclear translocation of Smad3, similar to effects observed in renal fibroblasts [28,33]. Notably, PKG phosphorylated Smad3 at different sites than activated ALK5 [33]. This suggests that the specific phosphorylation site determines whether Smad3’s nuclear translocation is induced or impeded [33]. Furthermore, the inhibition of cardiac fibrosis with Omega-3 fatty acids and chlorogenic acid treatment was also associated with PKG-mediated reduced nuclear translocation of Smad2/3 [34,35]. In both studies, these substances enhanced endothelial nitric oxide synthase (eNOS) activity and increased NO production, implicating the NO–sGC–cGMP pathway activation as the driver of the observed antifibrotic effects [34,35]. Additionally, in a pressure-induced model of heart failure, sGC stimulators mitigated cardiac fibrosis by inhibiting the expression and activity of angiotensin-converting enzyme (ACE), thereby reducing angiotensin II production in the myocardium [22]. The consequent reduction in angiotensin type 1 receptor activation decreased TGF-β expression, providing an additional antifibrotic mechanism for sGC stimulators.

Notably, the antifibrotic effects of NO–sGC–cGMP pathway modulation have also been demonstrated with vericiguat. In a mouse model, vericiguat attenuated angiotensin II–induced reactive oxygen species generation and downregulated genes associated with both cardiac fibrosis and hypertrophy. Histologic analysis confirmed a reduction in myocardial fibrotic area and decreased cardiomyocyte cross-sectional area. These cardioprotective effects appeared to be mediated, at least in part, by ERK signalling inhibition, underscoring the mechanistic overlap between the pathways driving fibrosis and hypertrophy and the corresponding antifibrotic and antihypertrophic actions of sGC stimulation [36].

## 4. WNT/β-Catenin and Hippo Signalling

The WNT/β-Catenin signalling pathway is a crucial regulator of cardiac fibroblast activity. Under physiological conditions, it plays a key role in restoring tissue integrity following injury. However, in the context of heart failure, its chronic activation promotes cardiac fibrosis [37]. In its inactive state, WNT/β-Catenin signalling is suppressed by a destruction complex composed of adenomatous polyposis coli (APC), Axin, casein kinase 1 (CK1), and glycogen synthase kinase 3 (GSK3) [38]. This complex binds β-Catenin, leading to a series of phosphorylations that ultimately result in its proteasomal degradation. In the presence of WNT ligands, activated membrane receptors Frizzled (FZD) and single-pass low-density lipoprotein receptor-related protein 5 or 6 (LRP5/6) sequester and inactivate the destruction complex. This allows β-Catenin to translocate to the nucleus, interact with other transcriptional factors, and induce profibrotic gene expression [38].

In cancer cells, enhancing PKG activity through PDE inhibitors has been shown to negatively regulate the WNT/β-Catenin pathway. Although the precise mechanisms are not yet fully understood, the inhibitory effects primarily target β-Catenin, reducing its expression, nuclear translocation, and transcriptional activity (Figure 2) [39].

PKG-mediated activation of Hippo signalling is another cancer-related finding that could explain the antifibrotic effects of sGC stimulators [40]. In its inactive state, the primary signalling molecules of the Hippo pathway, Yes-associated protein (YAP) and transcriptional co-activator with PDZ-binding motif (TAZ), can freely translocate to the nucleus. There, they interact with TEA domain family transcription factors (TEAD1–4) to activate genes promoting fibroblast proliferation and differentiation into myofibroblasts [40,41]. Conversely, activation of Hippo signalling activates the serine/threonine kinases mammalian STE20-like (Mst1/2), leading to a series of phosphorylations that target YAP and TAZ for ubiquitination and subsequent proteasomal degradation [40,41].

In prostate cancer cells, activation of NO–sGC–cGMP signalling, whether through NO or PDE inhibitors, resulted in Hippo pathway activation. The proposed mechanism involves direct activation of Mst1/2 by PKG, which leads to the destabilization and degradation of TAZ (Figure 2) [42].

Importantly, existing data on the inhibitory effects of NO–sGC–cGMP activation on WNT/β-Catenin and Hippo signalling largely derive from studies on tumor cells. Further research is needed to determine whether these effects are also applicable to cardiac fibroblasts.

## 5. Antihypertrophic Effects

In response to increased workload, the myocardium adapts through cardiomyocyte hypertrophy. This involves increased protein production and sarcomere reorganization, allowing the heart to maintain function despite increased demand [43]. However, prolonged hemodynamic stress triggers maladaptive changes that eventually lead to cardiac dysfunction and heart failure. Although the signalling pathways mediating maladaptive remodeling are diverse, they all result in fetal gene reactivation in cardiomyocytes, a hallmark of pathological hypertrophy [44].

Similar to suppressing fibrosis, sGC stimulators mitigate excessive hypertrophy in response to pathologic stress. In preclinical settings, treatment with an sGC stimulator reduced left ventricular mass index and cardiomyocyte cross-sectional area compared to controls [36,45]. These effects were accompanied by improved systolic function, demonstrating that NO–sGC–cGMP signalling inhibits pathologic hypertrophy while maintaining compensatory adaptations.

## 6. Calcineurin/NFAT Signalling

One of the best-studied pathways associated with pathological hypertrophy is the calcineurin/nuclear factor of activated T-cells (NFAT) signalling pathway. Calcineurin, a cytosolic protein with serine/threonine phosphatase activity, plays a central role in regulating signal transduction. Upon activation by increased intracellular calcium levels, calcineurin dephosphorylates NFAT, allowing it to translocate to the nucleus and regulate gene expression [46].

In preclinical studies, activation of the NO–sGC–cGMP pathway was shown to inhibit calcineurin signalling [46,47,48]. Pretreatment with NO donors and cGMP analogues decreased the nuclear translocation and transcriptional activity of NFAT [46]. These inhibitory effects disappeared in cells with constitutively active calcineurin or after PKG inhibition, indicating that PKG acts upstream of calcineurin [46]. PKG phosphorylates and inactivates several calcium channels, including transient receptor potential canonical (TRPC) channels, calcium release-activated calcium channel protein 1 (ORAI1), and L-type calcium channels [46,47,48]. This lowers intracellular calcium levels and thereby inhibits calcineurin activity (Figure 3).

## 7. RGS Signalling

In contrast to calcineurin-mediated pathological remodelling, regulators of G protein signalling (RGS) exhibit cardioprotective effects by inhibiting pathological hypertrophy in response to mechanical and neurohumoral stress [49]. Their protective action lies in their ability to modulate signalling mediated by G protein-coupled receptors (GPCRs) that drive pathological remodeling, including angiotensin II, endothelin-1, and beta-adrenergic signalling [49]. Upon ligand binding, GPCRs induce a conformational change in the heterotrimeric G protein, increasing its affinity for GTP. This change prompts the dissociation of the G protein into α and βγ subunits, which then interact with various effector molecules to transmit intracellular signals. Over time, the GTPase activity of the α subunit hydrolyzes GTP to GDP, leading to the reassembly of the G protein and termination of the signal. RGS proteins significantly shorten the duration of signal transmission by accelerating GTP hydrolysis [49].

In animal smooth muscle cells, activation of PKG by NO donors or cGMP resulted in the phosphorylation of RGS2 and RGS4 [50,51]. Phosphorylated RGS proteins exhibited enhanced GTPase activity, indicating that PKG amplifies their inhibitory action over GPCR signalling. Additionally, phosphorylation was linked to the translocation of RGS proteins from the cytoplasm to the membrane, positioning them closer to G proteins (Figure 3) [50,51]. PKG-mediated activation of RGS4 also demonstrated antihypertrophic effects in mice cardiomyocytes stimulated by endothelin-1 [52]. However, in this study, PKG activation was achieved via particulate guanylate cyclase activation with atrial natriuretic peptide, rather than NO, which activates sGC. Due to the different intracellular localizations of these two cGMP-generating enzymes, they increase PKG activity in distinct cellular regions, potentially influencing different targets. The specific role of NO–sGC–cGMP-mediated regulation of RGS proteins in the observed antihypertrophic effects in cardiomyocytes thus remains to be clarified.

## 8. mTOR Signalling

The roles of RGS and calcineurin signalling in cardiac function are well-defined, with RGS being protective and calcineurin signalling proving harmful. However, the role of mammalian target of rapamycin (mTOR) signalling in regulating cardiomyocyte hypertrophy is more complex. On one hand, mTOR signalling is essential for myocardial growth during development and in response to physiological stimuli such as exercise, with its absence leading to dilated cardiomyopathy. On the other hand, chronic activation of mTOR signalling by pathological stressors leads to maladaptive cardiac remodeling and heart failure [53].

mTOR functions within the mTOR complex (mTORC), where it acts as a catalytic subunit with serine/threonine protein kinase activity. Other components of mTORC provide structural and regulatory functions, adapting mTOR activity based on the cell’s nutritional and energy status. This regulation promotes cardiomyocyte growth and metabolism when resources are plentiful and represses these processes when resources are scarce. mTOR signalling is modulated by various molecules, notably the Tuberous Sclerosis Complex (TSC). TSC accelerates the hydrolysis of GTP on Rheb (Ras homolog enriched in brain), a key activator of mTOR. This GTPase-activating function of TSC leads to the inactivation of Rheb and subsequent inhibition of mTOR signalling [54].

The activity of TSC in cardiomyocytes is modulated by various kinases, including PKG. Studies on both human and animal cardiomyocytes have demonstrated that PKG activation, induced by PDE inhibitors, results in the phosphorylation and enhanced activation of TSC (Figure 3) [55]. In experimental models, mice with a phosphorylation-silencing mutation in TSC developed significant hypertrophy and cardiac dysfunction under pressure overload. Conversely, mice treated with PDE inhibitors or those with a phosphorylation-mimicking mutation of TSC showed protection against these adverse effects. Notably, PKG activation and the phosphorylation-mimicking mutation of TSC did not impact the basal activity of mTOR, suggesting that augmentation of NO–sGC–cGMP signalling could suppress pathological hypertrophy while simultaneously maintaining the necessary mTOR activity for cardiomyocyte homeostasis [55].

## 9. Anti-Inflammatory and Immunomodulatory Actions

Heart failure is closely associated with systemic inflammation, characterized by elevated levels of proinflammatory cytokines and increased immune cell activity. Rather than being an “innocent bystander”, inflammation actively drives maladaptive ventricular remodeling, exacerbating heart failure and heightening the risk of major adverse cardiovascular events [56].

The activation of immune and inflammatory responses is orchestrated by a network of interconnected signalling pathways, most of which converge on the activation of the inducible transcription factor nuclear factor-κB (NF-κB). In its inactive state, NF-κB is sequestered in the cytoplasm by the inhibitory protein IκBα. Upon stimulation by various inflammatory cytokines and microbial products, corresponding receptors activate IκB kinases, leading to the phosphorylation, ubiquitination, and proteasomal degradation of IκBα. This releases NF-κB, allowing it to translocate to the nucleus, where it binds to specific DNA elements known as κB enhancers. This interaction induces the expression of multiple proinflammatory cytokines (e.g., IL-1, IL-2, IL-6, IL-8, IL-12, IL-18, and TNF-α) and other signalling molecules critical for the differentiation and activation of adaptive and innate immune cells [57].

Notably, certain proinflammatory cytokines, such as IL-1β and IL-18, are synthesized in their inactive pro forms and require additional post-translational modifications for activation. These modifications are mediated by multimeric protein complexes known as inflammasomes. Among the various inflammasome-forming proteins, the nod-like receptor protein 3 (NLRP3) is the most extensively studied. Upon activation by diverse signals, NLRP3 oligomerizes and recruits apoptosis-associated speck-like protein containing a CARD (ASC) and pro-caspase 1. This interaction triggers autoactivation of caspase 1, which is crucial for the maturation and activation of IL-1β and IL-18 [58].

Preclinical studies have highlighted the potent anti-inflammatory and immunomodulatory effects of augmenting the NO–sGC–cGMP signalling pathway. In human endothelial and animal macrophage cell cultures stimulated with TNF-α, pretreatment with NO donors resulted in reduced expression of NF-κB-regulated genes [59]. This inhibitory effect was accompanied by increased expression and nuclear translocation of IκBα, suggesting that NO extends the inhibitory activity of IκBα to the nucleus. Interestingly, subsequent studies revealed that these effects of NO are mediated independently of PKG. Instead, increased cGMP concentration was shown to inhibit PDE3, elevating cAMP levels and activating protein kinase A (PKA), which proved essential for NO-mediated inhibition of NF-κB signalling [60]. Conversely, in hepatic macrophage cells pretreated with sGC stimulators, NF-κB inhibition was linked to PKG-mediated phosphorylation of vasodilator-stimulated phosphoprotein (VASP) [61]. Although multiple studies have validated the anti-inflammatory effects of NO/VASP signalling, the precise mechanisms remain unknown. Additionally, sGC stimulators have been shown to disrupt NLRP3 inflammasome assembly, further contributing to their anti-inflammatory actions (Figure 4) [61].

Table 1 summarizes all presented preclinical studies on the antifibrotic, antihypertrophic, and anti-inflammatory effects of NO–sGC–cGMP modulation, together with the proposed mechanisms.

## 10. Clinical Trials of Vericiguat in Chronic Heart Failure

### 10.1. Large Randomized Controlled Trials

In 2015, the placebo-controlled, dose-finding SOCRATES-REDUCED trial provided the first clinical evidence for vericiguat in patients with HFrEF and recent worsening. The study enrolled 456 symptomatic patients with a left ventricular ejection fraction (LVEF) <45% and an episode of decompensation within the preceding month. Participants were randomized into five groups—matched for age, sex, comorbidities, and background heart failure therapy—to receive placebo or one of four target doses of vericiguat (1.25 mg, 2.5 mg, 5.0 mg, or 10.0 mg once daily). At 12 weeks, vericiguat produced a dose-dependent reduction in plasma NT-proBNP compared with placebo. Given the established prognostic value of NT-proBNP in heart failure, these biomarker changes provided the rationale for subsequent outcome trials [15].

The VICTORIA trial extended these findings by assessing whether such biomarker improvements translated into clinical benefit. A total of 5050 high-risk patients with HFrEF and recent worsening—defined as hospitalization or the need for intravenous diuretics within the preceding six months—were randomized to vericiguat or placebo on top of optimized guideline-directed medical therapy. Over a median follow-up of 10.8 months, vericiguat reduced the primary composite endpoint of cardiovascular death or first heart failure hospitalization (35.5% vs. 38.5%; hazard ratio [HR] 0.90; 95% confidence interval [CI] 0.82–0.98; *p* = 0.02). This benefit was mainly driven by fewer hospitalizations (1223 vs. 1336 events; HR 0.91; 95% CI 0.84–0.99; *p* = 0.02), while cardiovascular mortality alone was not significantly different. The number needed to treat to prevent one primary event was 24—comparable to other therapies in similar high-risk cohorts over a relatively short follow-up [16]. These results supported a Class IIb recommendation for vericiguat in the 2022 ACC/AHA and 2021 ESC heart failure guidelines [62,63].

Building on these results, the recently published VICTOR trial evaluated vericiguat in a more stable population of HFrEF patients with NT-proBNP levels below 6000 pg/mL and no episode of worsening heart failure within the previous six months. Nearly half had never been hospitalized for HF, and 79% were in New York Heart Association (NYHA) class II, reflecting an early and well-compensated disease stage. While the primary composite endpoint of cardiovascular death or first HF hospitalization was neutral, vericiguat significantly lowered all-cause mortality (12.3% vs. 14.4%; HR 0.84, 95% CI 0.74–0.97; *p* = 0.015) and cardiovascular mortality (9.6% vs. 11.3%; HR 0.83, 95% CI 0.71–0.97; *p* = 0.020). This benefit extended to both sudden cardiac death (2.7% vs. 3.6%; HR 0.75, 95% CI 0.56–0.99; *p* = 0.042) and HF-related deaths (2.9% vs. 4.0%; HR 0.71, 95% CI 0.54–0.94; *p* = 0.016). Notably, the reduction in cardiovascular mortality was evident as early as 11 months and persisted throughout the median follow-up of 19.7 months.

### 10.2. Smaller Mechanistic and Functional Studies

Several smaller studies have evaluated vericiguat in HFrEF. In an observational cohort of 103 patients with recent worsening, Galván Ruiz et al. reported improved quality of life over six months, measured by the EuroQol-5D index (0.83 to 0.87, *p* = 0.032) and visual analogue scale (60 to 79, *p* = 0.005). NYHA functional class also improved, with class III decreasing from 67.3% to 22.4% and class II increasing from 32.7% to 75.5% (*p* < 0.001). In the same cohort, annual rates of heart failure decompensations requiring hospitalization or intravenous diuretics fell from 2.3 ± 1.4 events to 0.79 ± 1.14 (*p* < 0.001), although the non-randomized design limits causal inference [64].

Zhan et al. found that adding vericiguat to standard therapy improved exercise capacity on cardiopulmonary exercise testing, with higher peak oxygen consumption, increased anaerobic threshold, and reduced ventilatory equivalent for carbon dioxide slope. These gains were accompanied by reductions in NT-proBNP and soluble ST2 (sST2) at all follow-up points, with significant changes seen as early as one month (NT-proBNP: 558.3 ± 147.8 vs. 512.6 ± 82.4, *p* = 0.017; sST2: 51.2 ± 17.4 vs. 42.7 ± 9.6, *p* = 0.001) [65]. Similarly, Tian et al. observed fewer patients with NT-proBNP > 1000 pg/mL in the vericiguat group than in controls (63.0% vs. 82.7%, *p* = 0.002), though allocation based on patient choice introduces the possibility of selection bias [66].

Collectively, these smaller mechanistic and functional studies reinforce vericiguat’s capacity to favorably modulate key biomarkers and exercise tolerance, albeit in less rigorously controlled settings.

Not all findings have been uniformly positive. In the VERICIDuAT study, Botero et al. reported a non-significant change in NT-proBNP after six months (median, 7055 vs. 5699; *p* = 0.966) [67]. Their cohort had a higher mean patient age than most prior studies (78 ± 11.2 years vs. 55–71 years), raising the possibility that treatment response may diminish with advanced age, greater comorbidity burden, or more advanced disease stage.

## 11. Evidence in HFpEF

Evidence in HFpEF is even less consistent. In SOCRATES-PRESERVED, vericiguat improved Kansas City Cardiomyopathy Questionnaire (KCCQ) scores but did not significantly reduce NT-proBNP, while in VITALITY-HFpEF, no quality-of-life benefit was observed [68,69]. These results have dampened enthusiasm for further evaluation of vericiguat in HFpEF patient cohort.

## 12. Safety Profile

Across studies, vericiguat has demonstrated a safety profile comparable to placebo, with symptomatic hypotension and mild gastrointestinal discomfort being the most frequent adverse effects and treatment discontinuation uncommon.

### 12.1. Linking Preclinical Mechanisms to Clinical Outcomes

Although clinical studies with vericiguat have not directly evaluated its impact on myocardial hypertrophy and fibrosis, indirect evidence can be drawn from echocardiographic data. In the VICTORIA echocardiographic substudy, 208 HFrEF patients receiving vericiguat showed no significant differences compared with 211 placebo-treated controls over 8 months in any assessed echocardiographic parameter [70]. Assessed parameters included LV mass index (LVMI), LV end-diastolic and end-systolic diameters, right ventricular (RV) end-diastolic inflow diameter, and the E/e′ ratio, with the latter serving as a surrogate for diastolic function and, indirectly, myocardial fibrosis. Similarly, Tian et al. observed no significant changes in echocardiographic parameters when vericiguat was added to guideline-directed medical therapy (GDMT) compared with GDMT alone [66].

By contrast, Zhan et al. reported significant reductions in left ventricular end-diastolic and end-systolic dimensions, as well as RV inflow diameter, in patients receiving vericiguat on top of GDMT, suggesting additional favourable structural remodeling [65]. In line with these findings, our single-center, non-randomized clinical study also observed significant improvements in myocardial structure and function after 6 months of vericiguat added to GDMT [71]. Most notably, we found evidence of reverse remodeling of the right ventricle, with a reduction in tricuspid regurgitation gradient (35 ± 10 mmHg vs. 29 ± 8 mmHg; *p* = 0.04) and improved systolic function, as reflected by tricuspid annular plane systolic excursion (18.5 ± 4.2 mm vs. 21.3 ± 4.7 mm; *p* = 0.03). These findings suggest that vericiguat may address right-sided heart failure, which remains insufficiently addressed by current therapeutic strategies. However, E/e′ did not change significantly, which may be partly explained by only mildly elevated baseline diastolic dysfunction and the relatively short observation period, leaving limited room for measurable improvement. To more precisely characterize the antifibrotic and antihypertrophic effects of vericiguat, future studies should ideally use cardiac magnetic resonance imaging, which allows accurate quantification of both myocardial fibrosis content as well as cardiac muscle mass.

The divergence in findings may be attributable to differences in patient populations. While all trials enrolled HFrEF patients following recent worsening, those in the VICTORIA substudy were older (mean age 67.3 vs. 56.9 years) and more decompensated, as reflected by markedly higher baseline NT-proBNP levels (median 2816 pg/mL vs. mean 549.4 pg/mL in Zhan et al.) [65,66,70]. Advanced age and greater disease severity are both associated with heightened oxidative stress, which has mechanistic relevance for sGC stimulator efficacy. Under high oxidative stress, the haem iron within sGC can be oxidized to the ferric state, lowering NO binding affinity and sometimes causing haem loss. This process renders sGC unresponsive to vericiguat, which primarily acts by enhancing NO sensitivity through positive allosteric modulation. Consistent with this explanation, a VICTORIA trial subanalysis found a significant association between baseline NT-proBNP and vericiguat treatment effect, with patients in the lower two quartiles deriving the greatest benefit for the composite outcome of heart failure hospitalization or cardiovascular death [72]. The Tian et al. cohort, although younger on average (51 years), still had nearly half of the participants with NT-proBNP > 1000 pg/mL, aligning with this proposed hypothesis [66]. Another factor contributing to the observed discrepancies may be the variability in structural remodeling, with HF patients exhibiting varying degrees of hypertrophy and fibrosis. In those with minimal maladaptive remodeling, the mechanistic targets of vericiguat may be insufficiently expressed to yield measurable benefit, whereas in advanced disease, the remodeling process may have progressed too far for clinically meaningful reverse remodeling.

The VICTOR trial provided important evidence that vericiguat improves outcomes in a stable HFrEF population as well. Contrasting the VICTORIA data, vericiguat reduced both cardiovascular and all-cause mortality across baseline NT-proBNP levels, although eligibility was restricted to patients with NT-proBNP < 6000 pg/mL [17]. By comparison, secondary analyses of VICTORIA indicated benefit up to levels approaching 8000 pg/mL, suggesting that an interaction between baseline biomarker burden and treatment effect may still exist. Currently, mechanisms underlying the mortality benefit observed in VICTOR trial remain speculative. Based on the subgroup analyses, the benefit appeared consistent across demographic, clinical, and therapeutic patient profiles. While this broad effect is clinically encouraging, it also poses a challenge for future studies aiming to define the biological pathways through which vericiguat exerts its clinical benefits. As discussed in this review, NO–sGC–cGMP stimulation may confer antifibrotic, antihypertrophic, and anti-inflammatory effects in the myocardium while also influencing vascular and systemic processes. Incremental improvements across multiple domains could cumulatively translate into a measurable survival advantage, even if individual endpoints fail to capture these effects. Developing composite mechanistic indices that integrate biomarkers, imaging, and functional parameters may better clarify the pathways underlying clinical benefit and guide patient selection.

Considering the effects of vericiguat on systemic inflammation, a biomarker analysis of the SOCRATES-REDUCED trial reported a dose-dependent reduction in high-sensitivity CRP following 3-month treatment with vericiguat, with the group receiving the target dose of 10 mg displaying a significantly greater reduction compared to placebo (−31.9% vs. 0.2%; *p* = 0.035) [73]. Similarly, Cao et al. reported a significant decrease in circulating CRP levels in 55 patients receiving vericiguat compared with 55 controls [74]. Notably, this study only included subjects with hypertensive heart failure, suggesting that the anti-inflammatory effects of vericiguat may not be restricted to HFrEF but could also be applicable to patients with HFmrEF and HFpEF. In the VICTORIA biomarker substudy, however, treatment with vericiguat resulted in no significant changes in evaluated biomarkers, including hs-CRP and IL-6 [75].

These discrepancies cannot be fully explained by differences in patient age or baseline NT-proBNP, as values were comparable across studies. Instead, they may reflect the limitations of classifying heart failure solely by left ventricular ejection fraction. While reduced ejection fraction invariably triggers RAAS and sympathetic activation, this is not necessarily the case for other signalling pathways relevant to HF pathophysiology, including systemic inflammation and oxidative stress. A precision medicine approach—using detailed profiling of gene expression, protein modifications, and metabolic signatures—could help identify subgroups most likely to benefit from NO–sGC–cGMP modulation. Such stratification would not only improve patient selection for vericiguat therapy but also guide the development of novel agents targeting specific maladaptive pathways in chronic heart failure patient cohort.

### 12.2. Ischemic vs. Nonischemic Heart Failure

Differences in heart failure etiology may influence the therapeutic response to vericiguat. Ischemic heart disease, in particular, represents a mechanistically compelling target for NO–sGC–cGMP pathway modulation. Experimental studies link reduced sGC activity to several processes driving atherosclerosis progression, including neointimal formation and vascular smooth muscle cell proliferation [11,76]. Impaired cGMP signaling also upregulates endothelial adhesion molecules, which promote leukocyte recruitment, sustain vascular inflammation, and contribute to plaque vulnerability [77]. In addition, altered sGC signalling can shift platelet homeostasis toward a prothrombotic state, enhancing adhesion, aggregation, and activation [78,79].

Restoration of NO–sGC–cGMP signalling may therefore offer multiple benefits in ischemic HF. By improving endothelial function, supporting angiogenesis, and enhancing vasomotor responses, sGC stimulators such as vericiguat could help align myocardial blood flow with metabolic demand and potentially facilitate revascularization [11,80]. Vericiguat’s inhibitory effects on thrombus formation may also reduce major adverse cardiovascular events, a risk to which this patient population is particularly susceptible [81].

Despite these mechanistic considerations, there is currently no clinical evidence that patients with ischemic HF derive greater benefit than those with other etiologies. In a post hoc VICTORIA trial analysis, vericiguat reduced the primary composite endpoint irrespective of coronary artery disease (CAD) status [82]. Notably, these patients had recent decompensation and established, clinically significant CAD—defined by prior percutaneous coronary intervention, coronary artery bypass grafting, or myocardial infarction—settings in which the ischemic process was advanced and large portions of myocardium were already replaced by fibrotic scar. In such cases, revascularization or improved vasodilation may yield limited benefit. By contrast, patients with stable ischemic disease without extensive scar burden may represent a population in which NO–sGC–cGMP stimulation is most effective. In this setting, vericiguat could stabilize vulnerable plaques, reduce thrombus formation, support collateral vessel growth, and improve endothelial function—thereby slowing progression and aligning perfusion with myocardial demand. In the VICTOR trial, vericiguat’s beneficial effects extended to patients with ischemic HF, with no significant differences in either primary or secondary outcomes compared with other etiologies [17]. Given these mechanistic considerations, initiating vericiguat early in patients with stable coronary artery disease could help reduce the incidence of major adverse cardiovascular events, slow progression to heart failure, and improve prognosis. Dedicated studies in this population are warranted to test this hypothesis.

### 12.3. Limitations of the Existing Evidence

Much of the mechanistic understanding of how NO–sGC–cGMP modulation may attenuate fibrosis, inflammation, and myocardial hypertrophy derives from preclinical studies in animal models and cell cultures, many of which examined non-cardiac tissues. While these experiments provide valuable mechanistic hypotheses, extrapolation to human heart failure warrants caution, as both the magnitude and nature of NO–sGC–cGMP effects can vary across species and cell types.

In several of the studies included in this review, NO donors, PDE inhibitors, or cGMP analogues were used instead of sGC stimulators to modulate the NO–sGC–cGMP pathway. Although these agents act on the same signaling cascade, their pharmacodynamic properties differ. In addition to activating sGC, NO donors induce protein S-nitrosylation, which can alter protein function and may contribute to observed effects independently of NO–sGC–cGMP signaling. PDE inhibitors raise cGMP concentrations by slowing its breakdown. However, their isoform selectivity determines which intracellular cGMP pools are augmented, leading to effects that may differ from sGC-mediated cGMP generation. Moreover, some PDE inhibitors also increase cyclic adenosine monophosphate (cAMP) levels, thereby engaging additional signaling mechanisms unrelated to sGC stimulation.

Nevertheless, because the signaling pathways identified in these preclinical studies are also implicated in the pathogenesis of human heart failure, these pre-clinical findings provide a strong rationale for future clinical and translational research to clarify the underlying mechanisms and evaluate the therapeutic potential of sGC stimulators in this setting.

## 13. Conclusions

The NO–sGC–cGMP pathway is vital for cardiovascular homeostasis, and its disruption in heart failure leads to maladaptive ventricular remodelling. Pharmacologic augmentation of this pathway, particularly through sGC stimulators, offers a promising therapeutic strategy. Preclinical studies demonstrate consistent antifibrotic, antihypertrophic, and anti-inflammatory effects, while clinical trials confirm reductions in death and hospitalization. Yet the link between these biological effects and observed clinical outcomes remains incompletely defined, and much of the mechanistic evidence is derived from non-human models. Future research should focus on clarifying the human pathways involved—using biomarkers, imaging, and patient stratification—to identify subgroups most likely to benefit. By bridging preclinical insights with precision medicine approaches, sGC stimulation could expand and refine the therapeutic landscape for heart failure.

## Figures and Tables

**Figure 1 cells-14-01400-f001:**
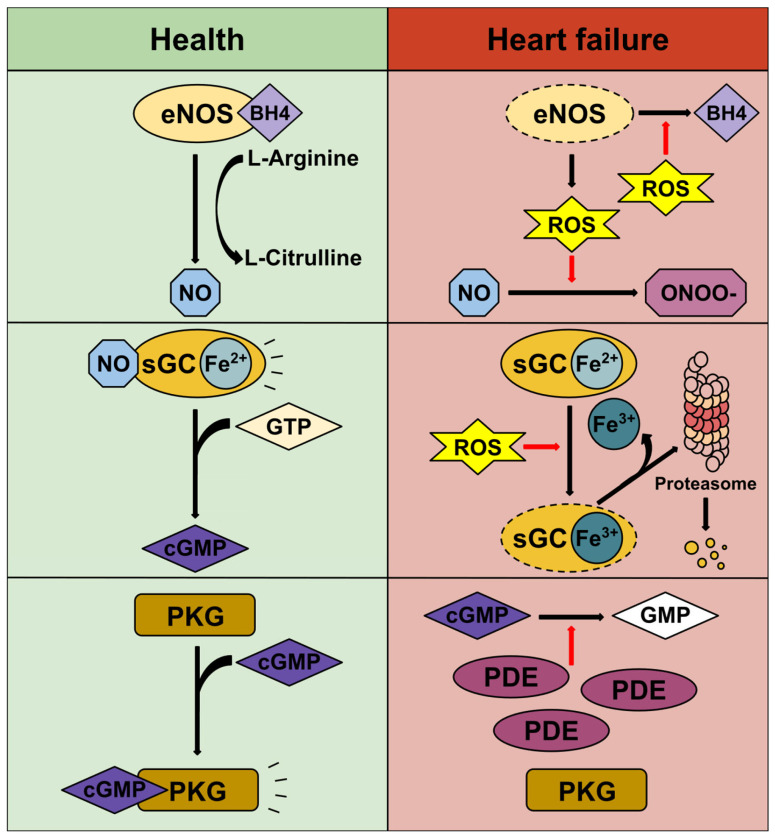
NO–sGC–cGMP signalling in health and disease. In heart failure, every stage of the NO–sGC–cGMP pathway is impaired. Oxidative stress induces endothelial nitric oxide synthase (eNOS) uncoupling, causing the enzyme to generate reactive oxygen species rather than nitric oxide (NO). Furthermore, any NO that is produced is rapidly oxidised to peroxynitrite (ONOO^−^), further reducing NO bioavailability. At the level of soluble guanylate cyclase (sGC), reactive oxygen species oxidise the ferrous (Fe^2+^) haem iron to its ferric (Fe^3+^) form, leading to haem loss and increased susceptibility to proteasomal degradation. Finally, the small amount of cyclic guanosine monophosphate (cGMP) that is generated under these conditions is degraded by upregulated phosphodiesterases, preventing it from activating downstream targets such as protein kinase G. eNOS, endothelial nitric oxide synthase; BH4, tetrahydrobiopterin; ROS, reactive oxygen species; NO, nitric oxide; ONOO^−^, peroxynitrite; sGC, soluble guanylate cyclase; GTP, guanosine triphosphate; cGMP, cyclic guanosine monophosphate; PKG, protein kinase G; GMP, guanosine monophosphate; PDE, phosphodiesterase.

**Figure 2 cells-14-01400-f002:**
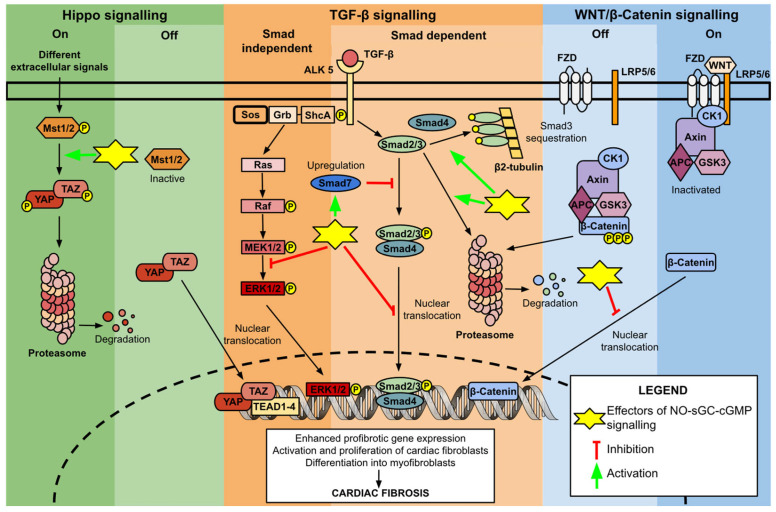
Schematic representation of the signalling pathways implicated in cardiac fibrosis. The scheme is divided into three sections: The green section represents Hippo signalling (light green deactivated state, dark green activated state); The orange section represents TGF-β signalling (dark orange Smad-independent signalling, light orange Smad-dependent signalling); The blue section represents WNT/β-Catenin signalling (light blue deactivated state, dark blue activated state). Yellow stars with corresponding pointers mark the antifibrotic mechanisms of NO–sGC–cGMP activation identified in preclinical studies. Mst1/2, serine/threonine kinases mammalian STE20-like; YAP, Yes-associated protein; TAZ, transcriptional co-activator with PDZ-binding motif; TEAD1-4, TEA domain family transcription factors; TGF-β, transforming growth factor beta; ALK 5, activin receptor-like kinase; ERK1/2, Extracellular Signal-Regulated Kinase; FZD, Frizzled; LRP5/6, single-pass low-density lipoprotein receptor-related protein 5 or 6; CK1, Casein kinase 1; APC, Adenomatous polyposis coli; GSK3, Glycogen synthase kinase 3.

**Figure 3 cells-14-01400-f003:**
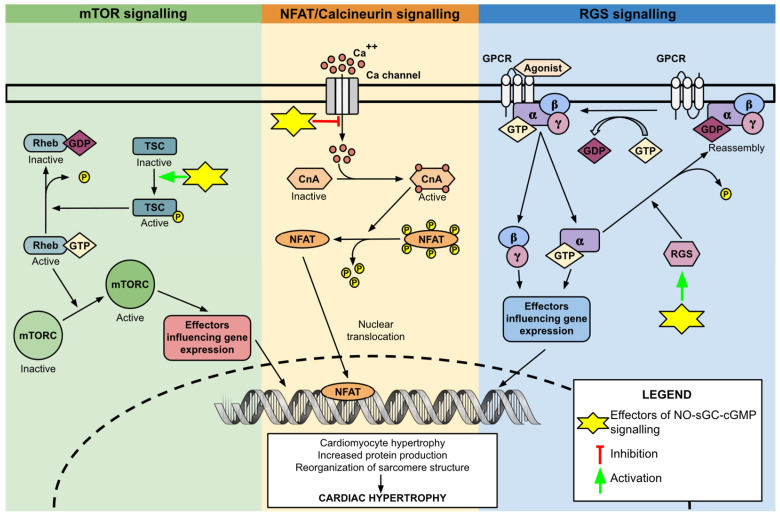
Schematic representation of the signalling pathways implicated in pathologic cardiac hypertrophy. The scheme is divided into three sections: The green section represents mTOR signalling; The yellow section represents NFAT/Calcineurin signalling; The blue section represents RGS signalling. Yellow stars with corresponding pointers mark the antihypertrophic mechanisms of NO–sGC–cGMP activation identified in preclinical studies. mTOR, mammalian target of rapamycin; GDP, guanosine triphosphate; GTP, guanosine diphosphate; TSC, Tuberous Sclerosis Complex; NFAT, nuclear factor of activated T-cells; CnA, Calcineurin; Ca++, Calcium ions; RGS, regulators of G protein signalling; GPCR, G protein-coupled receptors.

**Figure 4 cells-14-01400-f004:**
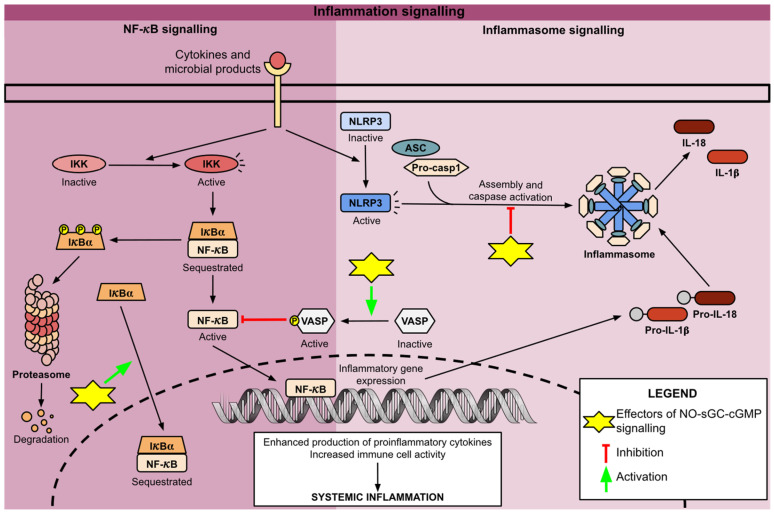
Schematic representation of the NF-κB and inflammasome signalling. The scheme is divided into two sections: The dark purple section represents NF-κB signalling; The light purple section represents inflammasome signalling. Yellow stars with corresponding pointers mark the anti-inflammatory and immunomodulatory mechanisms of NO–sGC–cGMP activation identified in preclinical studies. NF-κB, Nuclear factor-κB; IKK, IκB kinase; VASP, Vasodilator-stimulated phosphoprotein; NLRP3, Nod-like receptor protein 3; ASC, Apoptosis-associated speck-like protein containing a CARD; Pro-casp1, pro-caspase 1; IL-1β, Interleukin 1 beta; IL-18, Interleukin 18.

**Table 1 cells-14-01400-t001:** Summary of preclinical data on antifibrotic, antihypertrophic, and anti-inflammatory effects of nitric oxide–soluble guanylate cyclase–cyclic guanosine monophosphate (NO–sGC–cGMP) modulation, with reported mechanistic pathways. sGC, soluble guanylate cyclase; TGF-β, transforming growth factor beta; ERK, Extracellular Signal-Regulated Kinase; cGMP, cyclic guanosine monophosphate; PAI-1, plasminogen activator inhibitor-1; NASH, nonalcoholic steatohepatitis; ECM, extracellular matrix; NO, nitric oxide; NFAT, nuclear factor of activated T-cells; PDE, phosphodiesterase; TRPC6, transient receptor potential canonical channel 6; ANP, atrial natriuretic peptide; RGS4, regulator of G-protein signaling 4; mTOR, mammalian target of rapamycin; TSC2, tuberous sclerosis complex 2; NF-κB = nuclear factor kappa B; CNP = C-type natriuretic peptide; NLRP3, Nod-like receptor protein 3; VASP, Vasodilator-stimulated phosphoprotein.

First Author, Year	Model Type	Intervention	Pathways	Outcome
Antifibrotic effects				
Geschka et al., 2011 [21] **AC**	Dahl salt-sensitive rat with myocardial histology	sGC stimulator (Riociguat)	Not specified	Reduced expression of biomarkers of fibrosis; Reduced myocardial fibrosis
Masuyama, et al., 2009 [22] **AC**	Rat pressure-overload model with myocardial histology	sGC stimulator (BAY 41–2272)	Not specified	Reduced expression of biomarkers of fibrosis; Reduced myofibroblast transformation; Reduced myocardial fibrosis
Schinner et al., 2017 [28] **ANC**	Mouse renal fibrosis model with histology	sGC stimulator (BAY 41–8543)	TGFβ via ERK and Smad	Reduced expression of collagen and biomarkers of fibrosis
Beyer et al., 2015 [29] **HNC**, **ANC**	Human and murine fibroblast cell cultures	sGC stimulator (BAY 41–8543), cGMP analogue	TGFβ via ERK	Reduced fibroblast activation and collagen deposition
Gong et al., 2011 [30] **ANC**	Rat pulmonary artery smooth muscle cell cultures	cGMP analogue	TGFβ via sequestration of Smad in cytosol by β2-tubulin	Reduced PAI-1 expression
Hall et al., 2019 [31] **HNC**, **ANC**	Mouse NASH model with histology Human and animal stellate cell cultures	sGC stimulator (praliciguat)	TGFβ via Smad	Reduced expression of fibrotic biomarkers; Reduced liver fibrosis
Li et al., 2008 [33] **AC**	Mouse cardiac fibroblast cell cultures	cGMP analogue	TGFβ via Smad	Reduced expression of biomarkers of fibrosis; Reduced deposition of ECM; Reduced myofibroblast transformation
Harada et al., 2025 * [36] **AC**	Mice AngII-induced heart failure with myocardial histology	sGC stimulator (vericiguat)	TGFβ via ERK	Reduced myocardial fibrosis; Reduced expression of hypertrophy-related genes; Reduced cardiac hypertrophy
Antihypertrophic effects				
Fiedler et al., 2002 [46] **AC**	Rat cardiomyocyte cell culture	NO donors, cGMP analogues	Calcineurin/NFAT via L-type Ca channel	Suppressed cardiomyocyte hypertrophy
Koitabashi et al., 2010 [47] **AC**	Rat model of cardiac stress and cardiomyocyte culture	PDE inhibitors, cGMP analogues	Calcineurin/NFAT via TRPC6	Reduced cardiac hypertrophy
Wang et al., 2015 [48] **HC**	Human cardiomyocyte cell culture	NO donor, cGMP analogue	Calcineurin/NFAT via Orai1	Reduced cardiac hypertrophy
Tokudome et al., 2008 [52] **AC**	Murine model of cardiac hypertrophy and cardiomyocyte culture	ANP	RGS4	Reduced expression of hypertrophy-related genes; Reduced cardiac hypertrophy
Ranek et al., 2019 [55] **AC**	Murine model of pathological cardiac stress and cardiomyocyte culture	PDE inhibitors	mTOR via TSC2	Reduced cardiac hypertrophy
Anti-inflammatory/immunomodulatory effects				
Spiecker et al. 1997 [59] **HNC**, **ANC**	Human endothelial and murine macrophage-like cell co-cultures	NO donors	NF-κB via IκBα	Reduced endothelial expression of adhesion molecules
Aizawa et al., 2003 [60] **ANC**	Rat vascular smooth muscle cell cultures	NO donors, CNP	NF-κB via PDE3	Reduced expression of NF-κB-dependent genes
Flores-Costa et al., 2020 [61] **ANC**	Mouse NASH model with liver histology	sGC stimulator (praliciguat)	NF-κB via VASP	Reduced expression of NF-κB-dependent genes; Reduced constituents of the NLRP3 inflammasome; Reduced liver infiltration with inflammatory cells

Legend: HC Human cardiomyocytes; HNC Human non-cardiac cells; AC Animal cardiomyocytes; ANC Animal non-cardiac cells; * Study reports both antifibrotic and antihypertrophic effects.

## Data Availability

No new data were created or analyzed in this study. Data sharing is not applicable to this article.

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
