# Peer review of "Targeting the NO–sGC–cGMP Pathway: Mechanisms of Action of Vericiguat in Chronic Heart Failure"

_cells, 2025, doi:10.3390/cells14171400_

Round 1

Reviewer 1 Report

Comments and Suggestions for Authors

Targeting the NO–sGC–cGMP Pathway: Mechanisms of Action of Vericiguat in Chronic Heart Failure 

Major comments

According to Turnitin there is 

22% Overall Similarity
The combined total of all matches, including overlapping sources, for each…
Filtered from the report:

  • Bibliography

  • Quoted text

  • Cited text

  • Small matches (fewer than 8 words)

Regarding Figure 1, the authors state “reproduced from (3)”. Do they have permission to reproduce it?

Please add a paragraph discussing Heart Failure (HF) and Worsening Heart Failure (WHF), highlighting the pathophysiological significance and clinical implications of WHF as an emerging therapeutic target.

In addition, I recommend including a paragraph summarizing the latest clinical trials on vericiguat, such as the VICTORIA study and any recent data, to provide an updated perspective on its efficacy and safety in patients with HF and WHF.

Regarding the Discussion section, I suggest adding:

  • Results from new clinical trials, to provide an updated view of the evolving evidence.

  • A section on perspectives for individualized therapy, emphasizing precision medicine approaches in the management of heart failure.

  • References to additional recent publications, for example:

  •   doi: 10.1016/j.ejphar.2025.177670. doi: 10.1253/circj.CJ-24-0659. doi: 10.3390/jcm13144209. doi: 10.1002/ejhf.3501.  doi: 10.1007/s40256-024-00701-0

Reviewer 2 Report

Comments and Suggestions for Authors

This is a well-structured and timely review that synthesizes current knowledge regarding the NO–sGC–cGMP pathway, its disruption in heart failure, and the therapeutic potential of sGC stimulators such as vericiguat. The manuscript is thorough, well-referenced, and generally clear. However, I have some comments and suggestions that I believe could further strengthen your work:

- Add a focused paragraph on the potential role of vericiguat in patients with ischemic heart disease, highlighting both mechanistic rationale (endothelial dysfunction, oxidative stress) and any relevant clinical trial subgroup data.

- Briefly compare and contrast vericiguat with other guideline-directed medical therapies for HFrEF (e.g., ARNIs, SGLT2 inhibitors, MRAs), and discuss where sGC stimulators may fit within treatment algorithms or in combination therapy.

- Clearly distinguish which mechanistic insights are based on preclinical or non-cardiac models versus those validated in human cardiac tissue, and suggest future research directions to bridge these gaps.

- Provide a visual summary (figure or table) integrating the antifibrotic, antihypertrophic, and anti-inflammatory mechanisms discussed, linked to clinical outcomes where possible.

- Introduce a short section before the conclusion outlining key limitations of current evidence, especially the extrapolation from animal models to human clinical scenarios.

- Review the manuscript for overly complex or lengthy sentences and consider editing for greater clarity and flow.

- End the abstract and/or conclusion with a succinct summary of the main practical implications for clinicians and researchers.

Reviewer 3 Report

Comments and Suggestions for Authors

The manuscript entitled Targeting the NO–sGC–cGMP Pathway: Mechanisms of Action  of Vericiguat in Chronic Heart Failure is a narative review about the signalling pathways associated with maladaptive remodeling and heart failure progression that can be modulated by sGC stimulators, focusing on the antihypertrophic, antifibrotic, and anti-inflammatory effects of NO-sGC-cGMP signalling observed in preclinical studies.

The manuscript  is well written. The literature synthesis carried out by the authors is clear, concise and useful for preclinical studies.

I recommend to make a table with published preclinical data about antifibrotic, antihypertrophic, and anti-inflammatory and immunomodulatory actions; this can help researchers to find easier other way of research.

Round 2

Reviewer 1 Report

Comments and Suggestions for Authors

accepted